# Complex Time Approach to the Hamiltonian and the Entropy Production of the Damped Harmonic Oscillator

**DOI:** 10.3390/e27080883

**Published:** 2025-08-21

**Authors:** Kyriaki-Evangelia Aslani

**Affiliations:** Department of Mechanical Engineering, University of the Peloponnese, 26334 Patras, Greece; k.aslani@go.uop.gr

**Keywords:** QGT, arrow of time, complex time, Hamiltonian, entropy production, irreversible thermodynamics, Poisson brackets, GENERIC, DHO

## Abstract

The present work applies and extends the previously developed Quantitative Geometrical Thermodynamics (QGT) formalism to the derivation of a Hamiltonian for the damped harmonic oscillator (DHO) across all damping regimes. By introducing complex time, with the real part encoding entropy production and the imaginary part governing reversible dynamics, QGT provides a unified geometric framework for irreversible thermodynamics, showing that the DHO Hamiltonian can be obtained directly from the (complex) entropy production in a simple exponential form that is generalized across all damping regimes. The derived Hamiltonian preserves a modified Poisson bracket structure and embeds thermodynamic irreversibility into the system’s evolution. Moreover, the resulting expression coincides in form with the well-known Caldirola–Kanai Hamiltonian, despite arising from fundamentally different principles, reinforcing the validity of the QGT approach. The results are also compared with the GENERIC framework, showing that QGT offers an elegant alternative to existing approaches that maintains consistency with symplectic geometry. Furthermore, the imaginary time component is interpreted as isomorphic to the antisymmetric Poisson matrix through the lens of geometric algebra. The formalism opens promising avenues for extending Hamiltonian mechanics to dissipative systems, with potential applications in nonlinear dynamics, quantum thermodynamics, and spacetime algebra.

## 1. Introduction

The arrow of time—the observed asymmetry between past and future—remains one of the most profound questions in physics. While the fundamental laws governing classical and quantum mechanics are time-reversible [1], macroscopic phenomena clearly exhibit an irreversible temporal direction, manifested in processes such as the increase in entropy. This asymmetry has been at the center of many philosophical and scientific quests since the 19th century, with roots in the second law of thermodynamics, formulated by Clausius, and its statistical foundation developed by Boltzmann, who famously introduced the H-theorem to quantify entropy production [2,3].

At the heart of the time directionality paradox lies the connection between microscopic time-symmetric dynamics (e.g., Newtonian, Lagrangian, Hamiltonian, or Schrödinger evolution) and the irreversibility observed in macroscopic systems [4]. The standard explanation tells us that the universe began with low-entropy initial conditions, but this does not explain the fundamental encoding of temporal directionality in physical laws. Several physical frameworks—such as non-equilibrium statistical mechanics, thermodynamic geometry, and information theory—have tried to explain the arrow of time in terms of entropy production and causal emergence [5,6].

In the seminal work of Ilya Prigogine [7,8], the role of irreversibility and the arrow of time in natural phenomena are explored, particularly in far-from-equilibrium systems. He challenged the traditional reversible nature of time in physics and underlined how irreversibility is fundamental to understanding complex systems like those found in biology and far-from-equilibrium thermodynamics. He also discussed self-organization, a process where complex systems spontaneously develop order and structure without external direction. This process is intrinsically linked to the arrow of time and the non-equilibrium conditions that allow for such organization. It is evident that time is not just a mathematical parameter but a dynamic and active element in the universe, shaping the evolution and complexity of systems.

One of the most intriguing mathematical attempts to address temporal directionality is through the introduction of complex time. Unlike real-valued Newtonian or relativistic time, complex time t∈C incorporates an imaginary component. While originally introduced for mathematical convenience—such as in Wick rotations t→−iτ in quantum field theory—complex time has acquired an important role in the study of temporal asymmetry [9,10]. In this manner, complex time paths are essential in quantum tunneling for describing transitions across classically forbidden regions. The instanton method developed by Callan and Coleman uses imaginary time to compute quantum tunneling amplitudes, effectively encoding irreversible probability flow in a time-asymmetric trajectory [11]. In black hole thermodynamics and quantum cosmology, the work of Stephen Hawking and James Hartle revealed that complex time plays a fundamental role in defining the quantum state of the universe. The no-boundary proposal, which replaces singular temporal origins with smooth Euclidean geometries, illustrates how imaginary time can regularize singularities and encode time asymmetry [12,13,14].

One of the most recent attempts to integrate the concept of complex time within classical mechanics and irreversible thermodynamics is Quantitative Geometrical Thermodynamics (QGT). QGT is a new framework originally derived by Parker and Jeynes [15,16] that treats the complex entropy production as the Wick-rotated complex conjugate of the complex Hamiltonian using the complex time approach. Specifically, an analytical relation of the complex sum of the system’s Hamiltonian and entropy production (rate of entropy increase) is derived based on the consideration of causality as a Hilbert transform. In natural units, when complexified, one is the Wick-rotated complex conjugate of the other. Within this framework, Hamiltonian expressions can be derived for dissipative systems, which was not standard in physics until now. QGT has been already applied to explain the existence and the form of various natural structures like DNA, spiral galaxies, alpha particles, and black holes [15,16,17,18,19].

As discussed in the previous paragraph, one very important concrescence of QGT is the derivation of Hamiltonian structures for dissipative systems from the entropy production using complex time. But why is this useful? In general, the derivation of Hamiltonians for dissipative systems extends the powerful formalism of classical mechanics to real-world systems exhibiting irreversible behavior, such as friction, viscosity, and heat conduction. While traditional Hamiltonian mechanics is inherently conservative and time-reversible, many physical systems experience energy dissipation and entropy production. Embedding dissipative dynamics within a generalized Hamiltonian framework allows for a unified treatment of reversible and irreversible processes using geometric and algebraic tools such as Poisson brackets and symplectic structures [20,21]. This approach not only provides a deeper insight into the physical structure of dissipative phenomena but also enhances the modeling of complex systems in fields ranging from statistical mechanics and biophysics to control theory and numerical simulation. Moreover, it bridges microscopic conservative dynamics with macroscopic non-equilibrium behavior, enabling structure-preserving methods and compatibility with quantum and statistical formulations [22,23,24].

Beyond QGT, the extension of Hamiltonians in dissipative systems has been also explored by other frameworks, such as GENERIC (General Equation for Non-Equilibrium Reversible–Irreversible Coupling), the Caldirola–Kanai (CK) Hamiltonian, and the Bateman dual-system Hamiltonian. GENERIC combines the Poisson bracket formulation governing reversible systems with a symmetric, positive semi-definite dissipation bracket that ensures entropy production in accordance with the second law of thermodynamics. The entropy functional, which is traditionally a thermodynamic scalar, acts as a generator of irreversible phenomena, analogous to how the Hamiltonian generates reversible motion [25,26,27]. While GENERIC offers this dual-generator approach, QGT combines both Hamiltonian and entropy production into a single complex relation, offering a minimal yet thermodynamically consistent representation. These two frameworks, although conceptually distinct, converge in their treatment of irreversibility as a geometric phenomenon embedded within the evolution equations.

Considering the other two frameworks for extending the Hamiltonian formalism in dissipative systems, i.e., the Caldirola–Kanai (CK) Hamiltonian, and the Bateman dual-system Hamiltonian, both are classical and elegant, yet they differ from QGT in a fundamental way. The CK formalism incorporates dissipation by introducing a time-dependent exponential factor into the system’s variables, resulting in an explicitly time-dependent Hamiltonian [28,29,30]. The Bateman dual system enlarges the phase space by introducing an auxiliary degree of freedom with negative damping, such that the total system is conservative and the Hamiltonian remains time-independent [31,32,33]. While both models are standard methods to derive Hamiltonians for dissipative systems, QGT achieves this in a fundamentally different manner: it treats Hamiltonian and entropy production as Wick-rotated complex conjugates within a complexified time framework. In this context, deriving the Hamiltonian of a dissipative system from its entropy production emerges naturally from the geometry of QGT.

One paradigmatic test case (while also useful for laboratory experiments) of dissipative systems is the damped harmonic oscillator (DHO), whose irreversible dynamics has been analyzed through multiple lenses. Chandrasekar et al. [34] employed the modified Prelle–Singer method to derive explicit time-independent integrals of motion for the DHO across all damping regimes. These integrals enabled the derivation of Lagrangian and Hamiltonian formulations, offering a conservative description for the DHO. The authors further showed that the canonical equations reproduce the standard dynamics, and they suggested quantization schemes in momentum space. McDonald [35] showed that the DHO, despite its dissipative nature, can be formulated within the Hamiltonian framework by introducing a time-dependent canonical momentum, similar to the Caldirola–Kanai Hamiltonian. This construction preserves Liouville’s theorem and reveals constants of motion through canonical transformations. In this context, a Hamiltonian was derived in exponential form, which can be potentially used for any particle subject to velocity-dependent damping. The treatment can be extended to relativistic cases and systems with time-dependent damping or forcing, providing a unified view of dissipative dynamics in classical and quantum contexts. Öttinger, in his paper [36], applied the GENERIC (General Equation for the Non-Equilibrium Reversible–Irreversible Coupling) framework to the DHO, offering a thermodynamically consistent formulation that distinguishes reversible and irreversible dynamics. By introducing entropy as a state variable and separating the dynamics into antisymmetric (Hamiltonian) and symmetric (dissipative) contributions, the GENERIC formalism captures both energy conservation and entropy production. This approach provides a powerful geometric and variational structure for describing dissipation within classical mechanics.

In this work, it is shown that the QGT formalism provides a suitable framework for deriving the DHO Hamiltonian directly from entropy production, applicable across all damping regimes. As discussed above, the QGT formalism itself is not introduced here for the first time; it was developed in earlier work by Parker and Jeynes [15,16,17,18,19] as a unifying thermodynamic–geometric approach. By interpreting entropy production as the imaginary component of a complex Hamiltonian, McDonald’s result is recovered and generalized to all possible damping cases. Moreover, the result is embedded within a consistent Poisson bracket structure, while its relationship to the results of the GENERIC framework is discussed. In this context, the geometric isomorphism between the antisymmetric Poisson operator and the imaginary axis of complex time is explored using geometric algebra. The implications of this approach extend beyond the DHO, offering a general methodology for incorporating Hamiltonian structures into dissipative systems, while also suggesting new directions for unifying classical mechanics and irreversible thermodynamics through the lens of complexified time.

## 2. Problem Setup and Solution

A damped harmonic oscillator (DHO) is similar to the simple harmonic oscillator, including a resistive (damping) force $F_{d}=-b\dot{x}$, proportional to velocity, which acts on a moving object. The following second-order differential equation commonly describes this type of system [35,36]:(1)$$m\ddot{x}+b\dot{x}+kx=0,$$
or(2)$$\ddot{x}+\alpha \dot{x}+\omega_{0}^{2}x=0,$$
where α=bm is the damping parameter and ω02=km is the angular frequency of the undamped simple harmonic oscillator. The general solution to Equation (2), using McDonald’s notation [35], is(3)$$x\left(t\right)=B_{1}e^{\frac{1}{2}\left(a^{\prime }-\alpha \right)t}+B_{2}e^{-\frac{1}{2}\left(a^{\prime }+\alpha \right)t},$$
where $a^{\prime }=\sqrt{\alpha^{2}-4\omega_{0}^{2}}$, $B_{1}=\frac{{\left(a^{\prime }+\alpha \right)x}_{0}+2\upsilon_{0}}{2a^{\prime }}$, and $B_{2}=\frac{{\left(a^{\prime }-\alpha \right)x}_{0}-2\upsilon_{0}}{2a^{\prime }}$. By differentiating $x\left(t\right)$ over time, the velocity is derived as follows:(4)$$\dot{x}\left(t\right)=\upsilon \left(t\right)=C_{1}e^{\frac{1}{2}\left(a^{\prime }-\alpha \right)t}+C_{2}e^{-\frac{1}{2}\left(a^{\prime }+\alpha \right)t},$$
where $C_{1}=-B_{1}\frac{a^{\prime }+\alpha }{2}$ and $C_{2}=B_{2}\frac{a^{\prime }-\alpha }{2}$. The constants $B_{1}$ and $B_{2}$ (and consequently, $C_{1}$ and $C_{2}$) directly depend on the initial conditions of the problem, $x_{0}$ and $\upsilon_{0}$, where $x\left(0\right)=x_{0}$ and $\dot{x}\left(0\right)=\upsilon_{0}$.

The damped harmonic oscillator is distinguished into three characteristic cases depending on the value of $a^{\prime }=\sqrt{\alpha^{2}-4\omega_{0}^{2}}$, i.e., the overdamped motion for $a^{\prime }>0$, the underdamped motion for $a^{\prime }<0$, and the critically damped motion for $a^{\prime }=0$.

### 2.1. Overdamped Oscillator $\sqrt{\alpha^{2}-4\omega_{0}^{2}}>0$

If b is large enough for the inequality $\alpha >\omega_{0}$ to hold, then $\sqrt{\alpha^{2}-4\omega_{0}^{2}}>0$ and $a^{\prime }$ takes real values. Equations (3) and (4) describe the solutions of both position and velocity. In this case, the oscillator is characterized by a decaying amplitude depending on the initial conditions. There is no oscillatory motion; the oscillator slowly moves monotonically to zero, as shown in Figure 1. The amplitude decays away with a time constant that is *longer* than $2/\left(\alpha -a^{\prime }\right)$.

### 2.2. Critically Damped Oscillator $\sqrt{\alpha^{2}-4\omega_{0}^{2}}=0$

This is the limiting case where $\alpha^{2}=4\omega_{0}^{2}$. For this case, the solution of the position takes the form(5)$$x\left(t\right)=\left(B_{1}+tB_{2}\right)e^{-\frac{\alpha }{2}t},$$
where, similar to Equation (3), the constants $B_{1}$ and $B_{2}$ directly depend on the initial conditions of the problem, $x_{0}$ and $\upsilon_{0}$. This motion is also non-sinusoidal and evolves monotonically to zero, as shown in Figure 2. The critically damped solution goes to zero with the shortest time constant τd=2α, that is, the largest angular frequency.

### 2.3. Underdamped Oscillator $\sqrt{\alpha^{2}-4\omega_{0}^{2}}<0$

In this case, $a^{\prime }$ is purely imaginary, and the solution of the position reduces to(6)$$x\left(t\right)=e^{-\frac{\alpha }{2}t}\left(B_{1}\cos\omega_{1}t+B_{2}\sin\omega_{1}t\right),$$
where $\omega_{1}=\sqrt{\omega_{0}^{2}-{\left(\frac{\alpha }{2}\right)}^{2}}$. It is easy to see that Equation (6) is very similar to the underdamped case $x\left(t\right)=B_{1}\cos\omega_{0}t+B_{2}\sin\omega_{0}t$. The key differences are that the frequency is smaller $\left(\omega_{1}<\omega_{0}\right)$ and there is an overall exponential damping term $e^{-\frac{\alpha }{2}t}$. The motion is a damped sinusoidal oscillation, as illustrated in Figure 3. Here, the oscillation amplitude decreases exponentially with a time *constant*
τd=2α. Using the initial conditions $x\left(0\right)=x_{0}$ and $\dot{x}\left(0\right)=0$, it can be easily seen that $B_{1}=x_{0}$ and $B_{2}=\frac{\alpha x_{0}}{2\omega_{1}}$.

## 3. Hamiltonian Mechanics and Poisson Brackets

Hamiltonian mechanics provides a geometric formulation of classical and quantum mechanics where the state of a system is described by canonical coordinates of position and momentum $\left(x,p\right)$ in phase space, where $p=m\upsilon^{2}$. The evolution of the system is governed by a scalar function $H\left(x,p\right)$, the Hamiltonian, which typically corresponds to the total mechanical energy. More generally, the Hamiltonian H is defined as follows:(7)$$H\left(x,p\right)=T\left(p\right)+V\left(x\right),$$
where $T\left(p\right)$ is the kinetic energy and is $V\left(x\right)$ the potential energy of the system. For any particle of mass m in a potential $V\left(x\right)$, this becomes(8)$$H\left(x,p\right)=\frac{p^{2}}{2m}+V\left(x\right).$$

The equations of motion are given by Hamilton’s equations:(9)$$\dot{x}=\left\{x,\;H\right\}=\frac{\partial H}{\partial p},\;\dot{p}=\left\{p,\;H\right\}=-\frac{\partial H}{\partial x}\;.$$These can be compactly expressed using the Poisson bracket $\left\{\cdot ,\cdot \right\}$, defined for any two functions $A\left(x,p\right)$ and $B\left(x,p\right)$ as follows:(10)$$\left\{A,\;B\right\}=\frac{\partial A}{\partial x}\frac{\partial B}{\partial p}-\frac{\partial A}{\partial p}\frac{\partial B}{\partial x}.$$The evolution of any observable $A\left(x,p\right)$ is then given by(11)$$\frac{dA}{dt}=\left\{A,\;H\right\}.$$

This structure ensures that phase-space volume is preserved during evolution (Liouville’s theorem), and that the Hamiltonian is conserved. In this sense, the Poisson bracket formalism not only generates the equations of motion but also encodes the geometric properties of conservative dynamics. However, this framework applies strictly to conservative systems. Dissipative systems, such as those involving friction or damping, fall outside this traditional scope, since energy is dissipated and entropy is produced. To describe irreversible processes, a more general framework is needed—one that incorporates both reversible (Hamiltonian) and irreversible (dissipative) dynamics in a consistent way, such the GENERIC framework or Quantitative Geometrical Thermodynamics (QGT).

## 4. Irreversible Thermodynamics and Entropy Production

In a damped harmonic oscillator, the total mechanical energy (the sum of kinetic and potential energy) is not conserved, due to the energy being dissipated as heat or another form of energy through the damping mechanism. The instantaneous mechanical energy $U\left(t\right)$ is given by(12)$$U\left(t\right)=\frac{1}{2}m{\dot{x}}^{2}+\frac{1}{2}kx^{2}.$$The energy dissipation rate is derived by taking the time derivative of $U\left(t\right)$:(13)$$\frac{dU}{dt}=m\dot{x}\ddot{x}+kx\dot{x}=\dot{x}\left(m\ddot{x}+kx\right).$$Using the equation of motion (Equation (1)), we get(14)$$m\ddot{x}+kx=-b\dot{x}.$$Substituting into Equation (13) yields(15)$$\frac{dU}{dt}=\dot{x}\left(-b\dot{x}\right)=-b{\dot{x}}^{2}<0.$$This energy is dissipated as heat into the environment, implying entropy production. If the surrounding reservoir is at a constant temperature T, the rate of entropy production is(16)$${\dot{S}}_{prod}=\frac{dS}{dt}=\frac{1}{T}\left(-\frac{dU}{dt}\right)=\frac{b}{T}{\dot{x}}^{2}.$$Using the definition of momentum $p=m\dot{x}$, the rate of entropy production for the damped harmonic oscillator can be written as follows:(17)$${\dot{S}}_{prod}=\frac{b}{m^{2}T}p^{2}=\frac{\alpha }{mT}p^{2}.$$

This irreversible entropy generation is incompatible with standard Hamiltonian mechanics, which assumes zero entropy production. Among the first attempts to unify conservative and dissipative dynamics was the GENERIC framework (General Equation for Non-Equilibrium Reversible–Irreversible Coupling), which extends the phase space to include both energy and entropy. In GENERIC, the state of the system is described by a set of variables z (which now includes position, momentum, and entropy) and evolves according to the following equation [26,27,36]:(18)$$\frac{dz}{dt}=L\left(z\right)\cdot \frac{\delta E}{\delta z}+M\left(z\right)\cdot \frac{\delta S}{\delta z}.$$
where $E\left(z\right)$ is the total energy functional; $S\left(z\right)$ is the total entropy functional; $L\left(z\right)$ is the Poisson operator, which is antisymmetric and encodes reversible dynamics; and $M\left(z\right)$ is the dissipative operator, which is symmetric, positive semi-definite, and encodes irreversibility. The GENERIC structure ensures the following:$L\left(z\right)\cdot \frac{\delta S}{\delta z}=0$ (entropy is not affected by reversible dynamics);$M\left(z\right)\cdot \frac{\delta E}{\delta z}=0$ (energy is not affected by dissipative dynamics).

By analogy with Hamiltonian mechanics, where the evolution of any observable $A\left(x,p\right)$ can be calculated using the Poisson brackets (Equation (10)), it is equivalent for $A\left(x,p,S\right)$ to write(19)$$\frac{dA}{dt}=\left\{A,\;H\right\}+\left[A,\;S\right],$$
where(20)$$\left\{A,H\right\}=\left(\frac{\delta A}{\delta z},L\left(z\right)\frac{\delta E}{\delta z}\right),$$
and(21)$$\left[A,\;S\right]=\left(\frac{\delta A}{\delta z},M\left(z\right)\frac{\delta S}{\delta z}\right),$$

The GENERIC framework imposes mathematical and physical constraints to ensure thermodynamic consistency. This theoretical structure can elegantly embed the irreversible thermodynamics of a dissipative system, such as the damped oscillator within a generalized Hamiltonian formalism, thereby enabling a consistent description of entropy production and conservative mechanics in a unified framework.

## 5. Quantitative Geometrical Thermodynamics (QGT)

As discussed in the Introduction, in QGT, the physical time is defined as a complex dynamical variable. Instead of treating the imaginary parameter τ as isomorphic to the real-time variable t, τ is defined as the analytic continuation t into the complex plane, as follows:(22)$$z\equiv i\left(t+i\tau \right)=-\tau +it.$$That is, the real and imaginary components of the complex time z are explicitly distinguished, such that t and τ appear independent of each other.

Following the introduction of the complex time z, Velazquez, Parker, and Jeynes introduced the concept of actio-entropy S as a holomorphic function of the classical action $S_{cl}$ and the thermodynamic entropy $S_{th}$ across complex time z (see Equations (16) and (17)), which explicitly combines action and entropy as follows [19]:(23)$$S=\frac{S_{th}}{2k_{B}}+i\frac{S_{cl}}{\hbar },$$
where $k_{B}$ is the Boltzmann constant and ℏ is the reduced Planck constant (h/2π). It is obvious from Equation (23) that S is dimensionless. Since S is holomorphic, Cauchy–Riemann equations are valid, as follows:(24)$$\frac{1}{2k_{B}}\frac{\partial S_{th}}{\partial \tau }=-\frac{1}{\hbar }\frac{{\partial S}_{cl}}{\partial t},$$(25)$$\frac{1}{2k_{B}}\frac{\partial S_{th}}{\partial t}=\frac{1}{\hbar }\frac{{\partial S}_{cl}}{\partial \tau }.$$Using the conventional definition of the Hamiltonian H, which is related to dissipationless (reversible) processes, and the entropy production Π (or ${\dot{S}}_{prod}$ from Equations (16) and (17)), related to dissipative (irreversible) processes, the following is derived:(26)$$H=-\frac{{\partial S}_{cl}}{\partial t},\;\Pi =\frac{\partial S_{th}}{\partial \tau }.$$
where t is associated with reversible processes, while τ is associated with irreversible processes.

When a function is holomorphic and the Cauchy–Riemann equations are satisfied, differentiating across the complex plane z is given by the Wirtinger operator, as follows:(27)$$\frac{\partial }{\partial z}=\frac{1}{2}\left(\frac{\partial }{\partial \tau }+i\frac{\partial }{\partial t}\right).$$In this way, the complex differential of the actio-entropy is evaluated as follows:(28)$$H_{z}=2i\frac{{\partial S}_{cl}}{\partial z}=-\frac{{\partial S}_{cl}}{\partial t}+i\frac{{\partial S}_{cl}}{\partial \tau }\equiv H+iH_{\tau },$$(29)$$\Pi_{z}=2\frac{{\partial S}_{th}}{\partial z}=\frac{{\partial S}_{th}}{\partial \tau }+i\frac{{\partial S}_{th}}{\partial t}\equiv \Pi +i\Pi_{t}.$$It should be emphasized that H represents the real (reversible) part of the complex Hamiltonian $H_{z}$ (the subscript z indicates the relation of the variable to the complex time z), while Π is regarded as the real (irreversible) part of the complex entropy production $\Pi_{z}$. The nature of any physical process, as described by the complex Hamiltonian $H_{z}$ and the complex entropy production $\Pi_{z}$, depends on the trajectory that the system follows through complex time. Specifically, the reversibility or irreversibility of the process (or any combination thereof) is determined by the combination of the real and imaginary components at each moment.

Within the framework of the temporal complex plane z, its counterpart (complex-conjugated) complex frequency plane is introduced as follows:(30)$$\hat{\omega }\equiv -i\left(\omega -iv\right)=-\left(v+i\omega \right).$$In this manner, the Fourier transform for the conjugate frequencies of the respective real and imaginary temporal components is defined:(31)$$F\left(\omega \right)=\int_{-\infty }^{\infty }f\left(t\right)e^{i\omega t}dt\to F\left(\hat{\omega }\right)=\int_{-\infty }^{\infty }f\left(z\right)e^{i\hat{\omega }z}dz.$$

Equation (31) shows that both time t and frequency ω are respectively analytically continued into their complex planes, with the complex time z and complex frequency $\hat{\omega }$ representing the appropriate conjugate pair, $\hat{\omega }⇋z$. For the case of the function H(t) being a causal (i.e., $H\left(t\right)=0$ for $t<t_{0}$, where $t_{0}$ is chosen as a convenient point in time to express the causality of the system) and also physically realizable (square-integrable) function, Cauchy’s theorem applies. This means that the Hamiltonian is holomorphic in the required (upper, as appropriate) half-plane, such that it obeys the dispersion relations (best known as practical constraints in signal processing). John Toll [37] gives rigorous proof that strict causality is logically equivalent to the existence of dispersion relations. Specifically, he shows that the real and imaginary properties of the dispersion are mutually related via the Kramers–Kronig relations, using the properties of the Hilbert transform. Following Toll‘s [37] Equation (2.5), the dispersion of the complex Hamiltonian (Equation (28)) using the component ω of the complex frequency $\hat{\omega }$ is written as follows:(32)$$H\left(\omega \right)=+\frac{1}{\pi }P\;\int_{-\infty }^{\infty }\frac{H_{\tau }\left(\omega^{\prime }\right)}{\omega^{\prime }-\omega }d\omega^{\prime },$$(33)$$H_{\tau }\left(\omega \right)=-\frac{1}{\pi }P\;\int_{-\infty }^{\infty }\frac{H\left(\omega^{\prime }\right)}{\omega^{\prime }-\omega }d\omega^{\prime },$$
where P is the principal part to be taken at the point $\omega^{\prime }=\omega$. It should be noted that because the real and imaginary parts of $H_{z}$ are Hilbert transforms of each other, they are indeed causal. Similarly, the dispersion of the complex entropy production (Equation (29)) using the component v of the complex frequency $\hat{\omega }$ is written as follows:(34)$$\Pi \left(v\right)=-\frac{1}{\pi }P\;\int_{-\infty }^{\infty }\frac{\Pi_{t}\left(v^{\prime }\right)}{v^{\prime }-v}dv^{\prime },$$(35)$$\Pi_{t}\left(v\right)=+\frac{1}{\pi }P\;\int_{-\infty }^{\infty }\frac{\Pi \left(v^{\prime }\right)}{v^{\prime }-v}dv^{\prime }.$$Again, P is the principal part to be taken at the point $v^{\prime }=v$. It is also noteworthy that applying the Hilbert transform to the negative temporal direction of the irreversible τ-axis reverses the conventional cause-and-effect relationship observed in the forward time direction. In other words, along the irreversible τ-axis, from the viewpoint of normal (forward-moving) time, the order of events is reversed: the “effect” precedes the “cause.”.

Using the mathematical properties of analytical continuation, one may write two symmetrical pairs of expressions for how the complex entropy production function and the complex Hamiltonian function relate along the two conjugate frequency axes forming the complex frequency plane:(36)$$\Pi_{t}\left(\omega \right)=-i\Pi \left(v\right),$$(37)$$\Pi \left(\omega \right)=i\Pi_{t}\left(v\right),$$(38)$$H_{\tau }\left(v\right)=-iH\left(\omega \right),$$(39)$$H\left(v\right)=iH_{\tau }\left(\omega \right).$$Using the Cauchy–Riemann equations (Equations (24) and (25)), it is now possible to relate the entropy production values on the real frequency v-axis to the appropriate Hamiltonian values on the imaginary frequency ω-axis.(40)$$\frac{1}{\hbar }H\left(\omega \right)=-i\frac{1}{2k_{B}}\Pi \left(v\right),$$(41)$$\frac{1}{\hbar }H\left(v\right)=i\frac{1}{2k_{B}}\Pi \left(\omega \right),$$(42)$$\frac{1}{\hbar }H_{\tau }\left(\omega \right)=i\frac{1}{2k_{B}}\Pi_{t}\left(v\right),$$(43)$$\frac{1}{\hbar }H_{\tau }\left(v\right)=-i\frac{1}{2k_{B}}\Pi_{t}\left(\omega \right).$$Using the formulation above, $H_{z}$ and $\Pi_{z}$ can be evaluated as follows:(44)$$\frac{1}{\hbar }H_{z}\left(\omega \right)=\frac{1}{\hbar }H\left(\omega \right)+i\frac{1}{2k_{B}}\Pi \left(\omega \right),$$(45)$$\frac{1}{2k_{B}}\Pi_{z}\left(v\right)=\frac{1}{2k_{B}}\Pi \left(v\right)-i\frac{1}{\hbar }H\left(v\right).$$In general, $H_{z}$ and $\Pi_{z}$ are evaluated as follows:(46)$$H_{z}\left(\omega \right)\equiv H\left(\omega \right)+iH_{\tau }\left(\omega \right),$$(47)$$H_{z}\left(v\right)\equiv H\left(v\right)+iH_{\tau }\left(v\right),$$(48)$$\Pi_{z}\left(v\right)\equiv \Pi \left(v\right)+i\Pi_{t}\left(v\right),$$(49)$$\Pi_{z}\left(\omega \right)\equiv \Pi \left(\omega \right)+i\Pi_{t}\left(\omega \right).$$Finally, in a slightly more compact form, $H_{z}$ and $\Pi_{z}$ are related as follows:(50)$$i{H_{z}}^{*}=\frac{\hbar }{2k_{B}}\Pi_{z}.$$The relevant (cross-axial) components of the complex Hamiltonian and entropy production in Equations (28) and (29) are also simplified as follows:(51)$$H_{z}=-\frac{{\partial S}_{cl}}{\partial t}+i\frac{{\partial S}_{cl}}{\partial \tau }\equiv H+i\frac{\hbar }{2k_{B}}\Pi ,$$(52)$$\Pi_{z}=\frac{{\partial S}_{th}}{\partial \tau }+i\frac{{\partial S}_{th}}{\partial t}\equiv \Pi +i\frac{2k_{B}}{\hbar }H.$$

## 6. Entropy Production and Hamiltonian of the Underdamped Harmonic Oscillator Using QGT

As discussed above, for conservative systems like an undamped harmonic oscillator, the Hamiltonian represents the system’s total energy and remains constant over time. On the other hand, damping introduces energy dissipation, which implies breaking time-reversal symmetry, making the system non-conservative. In such cases, the Hamiltonian is not given straightforwardly. Based on these, QGT can be employed to explore the dynamics of a DHO and perhaps derive a Hamiltonian from its entropy production. It should be noted that Parker and Jeynes [37] derived an analytic relation for the entropy production of a DHO using McDonald’s [37] Hamiltonian approach. Here, the opposite process is followed, where a Hamiltonian is derived from entropy production as it is given from irreversible thermodynamics.

As a start, the simplest case of the DHO will be considered, i.e., the underdamped oscillator where $\alpha \ll \omega_{0}$. The position of this oscillator as a function of time is given by Equation (6). Using generic boundary conditions $x\left(0\right)=x_{0}$ and $\dot{x}\left(0\right)=\upsilon_{0}$, Equation (6) is written as follows:(53)$$x\left(t\right)=e^{-\frac{\alpha }{2}t}\left[x_{0}\cos\omega_{1}t+\left(\frac{\upsilon_{0}+\frac{\alpha }{2}x_{0}}{\omega_{1}}\right)\sin\omega_{1}t\right],$$
while the velocity is calculated as follows:(54)$$\dot{x}\left(t\right)=\upsilon \left(t\right)=e^{-\frac{\alpha }{2}t}\left(C_{1}\cos\omega_{1}t+C_{2}\sin\omega_{1}t\right),$$
where $C_{1}=\upsilon_{0}$ and $C_{2}=-\frac{\alpha \upsilon_{0}}{2\omega_{1}}-\frac{\alpha^{2}x_{0}}{4\omega_{1}}-x_{0}\omega_{1}$.

Averaging over one period (see Appendix A), the total energy decays exponentially:(55)$$U\left(t\right)=U_{0}e^{-\alpha t},$$
where $U_{0}=\frac{1}{2}m\left({\upsilon_{0}}^{2}+{\omega_{0}}^{2}{x_{0}}^{2}\right)$, equal to the total mechanical energy of the undamped oscillator. Using Equation (55), $\frac{dU}{dt}$ is calculated as follows:(56)$$\frac{dU}{dt}=-\alpha U\left(t\right)=-\alpha U_{0}e^{-\alpha t}.$$In this manner, using Equation (16), ${\dot{S}}_{prod}$ is derived as follows:(57)$${\dot{S}}_{prod}=\frac{1}{T}\left(-\frac{dU}{dt}\right)=\frac{\alpha U_{0}}{T}e^{-\alpha t}.$$

Following the steps of Parker and Jeynes [15], the Fourier transform of ${\dot{S}}_{prod}$ is calculated using Equation (31) to determine its real and imaginary frequency components as follows:(58)$$\Pi_{z}\left(v\right)=\int_{-\infty }^{\infty }{\dot{S}}_{prod}e^{ivt}dt=\frac{\alpha U_{0}}{T}\int_{0}^{\infty }e^{-\alpha t}e^{ivt}dt=\frac{\alpha U_{0}}{T}\frac{1}{\alpha -iv},$$
where, without loss of generality, it is considered that ${\dot{S}}_{prod}=0$ for t<0. From Equation (58), it is evident that the complex entropy production $\Pi_{z}$ has real and imaginary frequency components:(59)$$\Pi_{r}\left(v\right)=\frac{\alpha U_{0}}{T}\frac{\alpha }{\alpha^{2}+v^{2}},$$(60)$$\Pi_{i}\left(v\right)=\frac{\alpha U_{0}}{T}\frac{v}{\alpha^{2}+v^{2}}.$$Similar to the Hamiltonian frequency components for the DHO in the paper of Parker and Jeynes [15], the two components of Equation (58) are also Hilbert transforms of each other, as causality requires. Using Equation (29) while including an additional instance of frequency α to maintain correct dimensionality, it is derived that(61)$$\Pi \left(v\right)=\frac{\alpha U_{0}}{T}\frac{\alpha^{2}}{\alpha^{2}+v^{2}},$$(62)$$\Pi_{t}\left(v\right)=\frac{\alpha U_{0}}{T}\frac{av}{\alpha^{2}+v^{2}}.$$From Equations (39) and (42), it is derived that(63)$$\Pi_{t}\left(v\right)=-\frac{2k_{B}}{\hbar }H\left(v\right).$$Using Equations (62) and (63), it is concluded that(64)$$H\left(v\right)=-\frac{\alpha U_{0}\hbar }{2k_{B}T}\frac{av}{\alpha^{2}+v^{2}}.$$From Equation (45), the complex entropy production associated with an underdamped oscillator is written as follows:(65)$$\frac{1}{2k_{B}}\Pi_{z}\left(v\right)=\frac{1}{2k_{B}}\Pi \left(v\right)-i\frac{1}{\hbar }H\left(v\right)\to \frac{1}{2k_{B}}\Pi_{z}\left(v\right)=\frac{\alpha }{2k_{B}T}\left[\frac{\alpha U_{0}}{\alpha^{2}+v^{2}}\left(a+iv\right)\right].$$

Equation (65) can be further simplified using the concept of relaxation time, i.e., the time needed for the underdamped oscillator to dissipate all of its energy, which in this case is τd=2α (see Section 2.3). As stated by Córdoba et al. [38], any nonstationary system thermalizes to its final equilibrium state in time of the order of Boltzmann time $\tau_{B}\;:=\;\frac{\hbar }{k_{B}T}$. It is possible to identify the oscillator’s mechanical relaxation time with Boltzmann time due to the fact that the oscillator is immersed in a thermal reservoir at temperature T and the damping is interpreted as arising from quantum–thermal dissipation (i.e., b∝ℏ). In this manner, relating the relaxation time of the underdamped oscillator with Boltzmann time, it is derived that(66)$$\tau_{d}=\frac{2}{\alpha }\equiv \frac{\hbar }{k_{B}T}\to T=\frac{\alpha \hbar }{2k_{B}}.$$It should be noted that Parker and Jeynes [15] derived a similar relation for temperature (see Equation (36) in their paper). Moreover, it should be also clarified that T denotes the constant temperature of the thermal reservoir, not the oscillator’s instantaneous mechanical temperature. It enters the formulation in the standard thermodynamic way, e.g., through entropy production $dS=\frac{dQ}{T}$ (see Equation (16)). The decay of the oscillator’s mechanical energy with time is a consequence of damping and does not require the reservoir temperature to change. In classical Brownian motion or fluctuation–dissipation descriptions, T is typically constant; the dissipated energy is transferred to the reservoir without lowering its temperature. In this manner, the identification $\tau_{d}\equiv \tau_{B}$ associates the constant damping rate α with the fixed bath temperature via $T=\frac{\alpha \hbar }{2k_{B}}$, without implying that T decreases as t→∞. It is also noted that, in contrast to the microscopic system–bath Hamiltonian framework, where integrating out bath degrees of freedom yields both dissipation and bath-induced fluctuations (with the fluctuation–dissipation theorem ensuring a finite-energy thermal equilibrium at a temperature T), the present QGT-based Hamiltonian is purely deterministic. As such, it models mechanical relaxation to the rest state rather than stochastic fluctuations about a finite-energy equilibrium. Extending the formalism to include fluctuation terms consistent with the fluctuation–dissipation theorem is an interesting avenue for future work, but it lies beyond the scope of the present study.

Inserting Equation (66) into Equation (65), the complex entropy production is now evaluated as follows:(67)$$\Pi_{z}\left(v\right)=\frac{2k_{B}}{\hbar }\left[\frac{\alpha U_{0}}{\alpha^{2}+v^{2}}\left(a+iv\right)\right],$$
while $\Pi \left(v\right)$ and $H\left(v\right)$ are calculated as follows:(68)$$\Pi \left(v\right)=\frac{2k_{B}U_{0}}{\hbar }\frac{\alpha^{2}}{\alpha^{2}+v^{2}},$$(69)$$H\left(v\right)=-U_{0}\frac{av}{\alpha^{2}+v^{2}}.$$

Using Equations (40) and (41), $H\left(\omega \right)$ and $\Pi \left(\omega \right)$ are derived from $\Pi \left(v\right)$ and $H\left(v\right)$ as follows:(70)$$H\left(\omega \right)=-i\frac{\hbar }{2k_{B}}\Pi \left(v\right),$$(71)$$\Pi \left(\omega \right)=-i\frac{2k_{B}}{\hbar }H\left(v\right).$$Inserting Equations (68) and (69) into Equations (70) and (71), it is concluded that(72)$$H\left(\omega \right)=-iU_{0}\frac{\alpha^{2}}{\alpha^{2}+\omega^{2}},$$(73)$$\Pi \left(\omega \right)=i\frac{2k_{B}U_{0}}{\hbar }\frac{a\omega }{\alpha^{2}+\omega^{2}}.$$From Equation (44), $H_{z}\left(\omega \right)$ is written as follows:(74)$$\frac{1}{\hbar }H_{z}\left(\omega \right)=\frac{1}{\hbar }H\left(\omega \right)+i\frac{1}{2k_{B}}\Pi \left(\omega \right)\to H_{z}\left(\omega \right)=i\frac{\alpha U_{0}}{\alpha^{2}+\omega^{2}}\left(a-i\omega \right).$$It is evident from Equations (67) and (74) that $i{H_{z}}^{*}=\frac{\hbar }{2k_{B}}\Pi_{z}$, as predicted from QGT (see Equation (50)). Moreover, Equations (72) and (73) are similar to Equation (32) from the paper of Parker and Jeynes [15]. Finally, the real Hamiltonian can now be derived using the inverse Fourier transform as follows:(75)$$H=\int_{-\infty }^{\infty }H_{z}\left(\omega \right)e^{-i\omega t}d\omega =\int_{0}^{\infty }i\frac{U_{0}}{\alpha^{2}+\omega^{2}}\left(a-i\omega \right)e^{-i\omega t}d\omega =U_{0}e^{\alpha t},$$
where the additional instance of frequency α is now subtracted. It is evident that Equation (75) is identical to Equation (9) from McDonald’s paper. In addition to McDonald’s methodology for deriving Hamiltonian formulations for dissipative systems, such as the DHO, the real Hamiltonian of the underdamped harmonic oscillator can also be predicted as a Wick-rotated complex conjugate of the entropy production using QGT. In this manner, the entropy production, as well as the Hamiltonian for the general solution of a DHO (a solution valid in all cases), will be calculated using QGT in the next section.

## 7. Entropy Production and Hamiltonian of the General Form of the DHO Using QGT

As can be seen from Section 2, a DHO is distinguished into three cases depending on the value of $a^{\prime }=\sqrt{\alpha^{2}-4\omega_{0}^{2}}$. Fortunately, these cases can be written in a closed form using a general solution that is valid in all of the parameter ranges and initial conditions (see Equation (3)). In this manner, the square of the velocity $\upsilon^{2}$ (see Equation (4)) is calculated as follows:(76)$$\upsilon^{2}\left(t\right)={C_{1}}^{2}e^{\left(a^{\prime }-\alpha \right)t}+2C_{1}C_{2}e^{-\alpha t}+{C_{2}}^{2}e^{-\left(a^{\prime }+\alpha \right)t}.$$Using the definition of entropy production (Equation (16)), the latter for the general form of the DHO is derived as follows:(77)$${\dot{S}}_{prod}=\frac{b}{T}\left({C_{1}}^{2}e^{\left(a^{\prime }-\alpha \right)t}+2C_{1}C_{2}e^{-\alpha t}+{C_{2}}^{2}e^{-\left(a^{\prime }+\alpha \right)t}\right).$$Following the steps from Section 6, the Fourier transform of ${\dot{S}}_{prod}$ is calculated using Equation (31) to determine its real and imaginary frequency components as follows:(78)$$\Pi_{z}\left(v\right)=\int_{-\infty }^{\infty }{\dot{S}}_{prod}e^{ivt}dt\;=\frac{b}{T}\left({C_{1}}^{2}\int_{0}^{\infty }e^{\left(a^{\prime }-\alpha \right)t}e^{ivt}dt+2C_{1}C_{2}\int_{0}^{\infty }e^{-\alpha t}e^{ivt}dt+{C_{2}}^{2}\int_{0}^{\infty }e^{-\left(a^{\prime }+\alpha \right)t}e^{ivt}dt\right)\to \;\Pi_{z}\left(v\right)=\frac{b}{T}\left(\frac{{C_{1}}^{2}}{\left(\alpha -a^{\prime }\right)-iv}+\frac{2C_{1}C_{2}}{\alpha -iv}+\frac{{C_{2}}^{2}}{\left(\alpha +a^{\prime }\right)-iv}\right),$$
where, for simplicity, it is considered that ${\dot{S}}_{prod}=0$ for t<0. From Equation (78), it is evident that the complex entropy production $\Pi_{z}$ has real and imaginary frequency components:(79)$$\Pi_{r}\left(v\right)=\frac{b}{T}\left({C_{1}}^{2}\frac{\alpha -a^{\prime }}{{\left(a^{\prime }-\alpha \right)}^{2}+v^{2}}+2C_{1}C_{2}\frac{\alpha }{\alpha^{2}+v^{2}}+{C_{2}}^{2}\frac{\alpha +a^{\prime }}{{\left(a^{\prime }+\alpha \right)}^{2}+v^{2}}\right),$$(80)$$\Pi_{i}\left(v\right)=\frac{b}{T}\left({C_{1}}^{2}\frac{v}{{\left(a^{\prime }-\alpha \right)}^{2}+v^{2}}+2C_{1}C_{2}\frac{v}{\alpha^{2}+v^{2}}+{C_{2}}^{2}\frac{v}{{\left(a^{\prime }+\alpha \right)}^{2}+v^{2}}\right).$$Similarly to the entropy frequency components for the underdamped harmonic oscillator (Equations (59) and (60)), the two components in Equations (79) and (80) are Hilbert transforms of each other, as causality requires. Using Equation (29) while including an additional instance of frequency α to maintain correct dimensionality, it is derived that(81)$$\Pi \left(v\right)=\frac{b}{T}\left({C_{1}}^{2}\frac{\alpha \left(\alpha -a^{\prime }\right)}{{\left(a^{\prime }-\alpha \right)}^{2}+v^{2}}+2C_{1}C_{2}\frac{\alpha^{2}}{\alpha^{2}+v^{2}}+{C_{2}}^{2}\frac{\alpha \left(\alpha +a^{\prime }\right)}{{\left(a^{\prime }+\alpha \right)}^{2}+v^{2}}\right),$$(82)$$\Pi_{t}\left(v\right)=\frac{b}{T}\left({C_{1}}^{2}\frac{\alpha v}{{\left(a^{\prime }-\alpha \right)}^{2}+v^{2}}+2C_{1}C_{2}\frac{\alpha v}{\alpha^{2}+v^{2}}+{C_{2}}^{2}\frac{\alpha v}{{\left(a^{\prime }+\alpha \right)}^{2}+v^{2}}\right).$$Using Equation (63) with Equation (82), it is concluded that(83)$$H\left(v\right)=-\frac{b^{2}\hbar }{2mk_{B}T}\left({C_{1}}^{2}\frac{v}{{\left(a^{\prime }-\alpha \right)}^{2}+v^{2}}+2C_{1}C_{2}\frac{v}{\alpha^{2}+v^{2}}+{C_{2}}^{2}\frac{v}{{\left(a^{\prime }+\alpha \right)}^{2}+v^{2}}\right).$$In this manner, the complex entropy production for the DHO $\Pi_{z}\left(v\right)$ is written as follows:(84)$$\Pi_{z}\left(v\right)=\frac{b}{T}\left({C_{1}}^{2}\frac{\alpha \left[\left(\alpha -a^{\prime }\right)+iv\right]}{{\left(a^{\prime }-\alpha \right)}^{2}+v^{2}}+2C_{1}C_{2}\frac{\alpha \left(a+iv\right)}{\alpha^{2}+v^{2}}+{C_{2}}^{2}\frac{\alpha \left[\left(\alpha +a^{\prime }\right)+iv\right]}{{\left(a^{\prime }+\alpha \right)}^{2}+v^{2}}\right).$$

As in the case of the underdamped harmonic oscillator (see Section 6), Equation (84) can be further simplified using the concept of relaxation time, i.e., Equation (66):(85)$$\Pi_{z}\left(v\right)=\frac{2mk_{B}}{\hbar }\left({C_{1}}^{2}\frac{\alpha \left[\left(\alpha -a^{\prime }\right)+iv\right]}{{\left(a^{\prime }-\alpha \right)}^{2}+v^{2}}+2C_{1}C_{2}\frac{\alpha \left(a+iv\right)}{\alpha^{2}+v^{2}}+{C_{2}}^{2}\frac{\alpha \left[\left(\alpha +a^{\prime }\right)+iv\right]}{{\left(a^{\prime }+\alpha \right)}^{2}+v^{2}}\right),$$
while $\Pi \left(v\right)$ and $H\left(v\right)$ are calculated as follows:(86)$$\Pi \left(v\right)=\frac{2mk_{B}}{\hbar }\left({C_{1}}^{2}\frac{\alpha \left(\alpha -a^{\prime }\right)}{{\left(a^{\prime }-\alpha \right)}^{2}+v^{2}}+2C_{1}C_{2}\frac{\alpha^{2}}{\alpha^{2}+v^{2}}+{C_{2}}^{2}\frac{\alpha \left(\alpha +a^{\prime }\right)}{{\left(a^{\prime }+\alpha \right)}^{2}+v^{2}}\right),$$(87)$$H\left(v\right)=-b^{2}\left({C_{1}}^{2}\frac{v}{{\left(a^{\prime }-\alpha \right)}^{2}+v^{2}}+2C_{1}C_{2}\frac{v}{\alpha^{2}+v^{2}}+{C_{2}}^{2}\frac{v}{{\left(a^{\prime }+\alpha \right)}^{2}+v^{2}}\right).$$

Using Equations (70) and (71), $H\left(\omega \right)$ and $\Pi \left(\omega \right)$ are derived from $\Pi \left(v\right)$ and $H\left(v\right)$ as follows:(88)$$H\left(\omega \right)=-im\left({C_{1}}^{2}\frac{\alpha \left(\alpha -a^{\prime }\right)}{{\left(a^{\prime }-\alpha \right)}^{2}+\omega^{2}}+2C_{1}C_{2}\frac{\alpha^{2}}{\alpha^{2}+\omega^{2}}+{C_{2}}^{2}\frac{\alpha \left(\alpha +a^{\prime }\right)}{{\left(a^{\prime }+\alpha \right)}^{2}+\omega^{2}}\right),$$(89)$$\Pi \left(\omega \right)=i\frac{2k_{B}b^{2}}{\hbar }\left({C_{1}}^{2}\frac{\omega }{{\left(a^{\prime }-\alpha \right)}^{2}+\omega^{2}}+2C_{1}C_{2}\frac{\omega }{\alpha^{2}+\omega^{2}}+{C_{2}}^{2}\frac{\omega }{{\left(a^{\prime }+\alpha \right)}^{2}+\omega^{2}}\right).$$From Equation (44), $H_{z}\left(\omega \right)$ is written as follows:(90)$$H_{z}\left(\omega \right)=-b\left({C_{1}}^{2}\frac{m\alpha \omega +i\left(\alpha -a^{\prime }\right)}{{\left(a^{\prime }-\alpha \right)}^{2}+\omega^{2}}+2C_{1}C_{2}\frac{\alpha \left(m\omega +i\right)}{\alpha^{2}+\omega^{2}}+{C_{2}}^{2}\frac{m\alpha \omega +i\left(\alpha +a^{\prime }\right)}{{\left(a^{\prime }+\alpha \right)}^{2}+\omega^{2}}\right).$$As was expected from the results of the underdamped oscillator, $i{H_{z}}^{*}=\frac{\hbar }{2k_{B}}\Pi_{z}$, a standard relation predicted from QGT (see Equation (50)). Finally, the real Hamiltonian can now be derived using the inverse Fourier transform (Equation (75)) as follows:(91)$$H=\int_{0}^{\infty }H_{z}\left(\omega \right)e^{-i\omega t}d\omega \to \;H=2me^{\alpha t}\left({C_{1}}^{2}\frac{\alpha }{a^{\prime }+\alpha }e^{\left(a^{\prime }-\alpha \right)t}+C_{1}C_{2}e^{-\alpha t}+{C_{2}}^{2}\frac{\alpha }{a^{\prime }-\alpha }e^{-\left(a^{\prime }+\alpha \right)t}\right).$$
where the additional instance of frequency α is now subtracted. Using $C_{1}=-B_{1}\frac{a^{\prime }+\alpha }{2}$ and $C_{2}=B_{2}\frac{a^{\prime }-\alpha }{2}$, the real Hamiltonian H can be written as follows:(92)$$H=\frac{m}{2}\;e^{\alpha t}\left({B_{1}}^{2}\alpha \left(a^{\prime }+\alpha \right)e^{\left(a^{\prime }-\alpha \right)t}+B_{1}B_{2}\left(\alpha^{2}-{a^{\prime }}^{2}\right)e^{-\alpha t}-{B_{2}}^{2}\alpha \left(a^{\prime }-\alpha \right)e^{-\left(a^{\prime }+\alpha \right)t}\right),$$
which coincides with the instantaneous mechanical energy $U\left(t\right)$ of the DHO multiplied by the factor $e^{\alpha t}$. This result matches that of the underdamped harmonic oscillator, with the difference that the mechanical energy U is not constant—an approach that was also adopted by Parker and Jeynes [15] in their own calculations. Furthermore, this result also coincides with McDonald’s expression for the Hamiltonian of the DHO (see Equation (9) in his paper), again with the distinction that the mechanical energy U does not remain constant. Finally, it is worth noting that Equation (93) coincides in form with the well-known Caldirola–Kanai Hamiltonian for the damped harmonic oscillator [30]. The final expression for the Hamiltonian of the general solution of the DHO is written as follows:(93)$$H=U\left(t\right)\;e^{\alpha t}=\left(\frac{1}{2}m{\dot{x}}^{2}+\frac{1}{2}kx^{2}\right)e^{\alpha t}.$$A detailed discussion of the comparison between Equation (93) and other known Hamiltonian formalisms for the DHO is provided in the Discussion.

To calculate the Poisson brackets for the underdamped harmonic oscillator, it is necessary to define the canonical momentum, also introduced by McDonald [35], rather than the mechanical momentum $p=m\dot{x}$:(94)$$p_{c}=m\dot{x}e^{\alpha t}=pe^{\alpha t}.$$The canonical momentum ensures that the system can be cast into a Hamiltonian form with a well-defined Poisson bracket structure that satisfies Liouville’s theorem. Physically, $p_{canonical}$ accounts for the energy loss introduced by the dissipative medium (in this case, the heat bath), scaling the momentum to embed the irreversible effects into the phase-space evolution. Moreover, the Hamiltonian of the general solution of the DHO, derived using QGT (Equation (93)), can be written in terms of the mechanical momentum p and the canonical momentum $p_{c}$:(95)$$H=U\left(t\right)\;e^{\alpha t}=\left(\frac{p^{2}}{2m}+\frac{kx^{2}}{2}\right)e^{\alpha t}=\frac{{p_{c}}^{2}}{2m}e^{-\alpha t}+\frac{kx^{2}}{2}e^{\alpha t}.$$In this manner, Hamilton’s equations (Equation (9)) are satisfied as follows:(96)$$\dot{x}=\left\{x,\;H\right\}=\frac{\partial H}{\partial p_{c}}=\frac{p_{c}}{m}e^{-\alpha t},\;\dot{p_{c}}=\left\{p_{c},\;H\right\}=-\frac{\partial H}{\partial x}=-kxe^{\alpha t}.$$By differentiating $\dot{x}$ one more time and using Equations (94) and (96), the DHO differential equation of motion is recovered (Equation (1)).

While the Hamiltonian formalism derived from QGT (see Equation (93)) embeds dissipative dynamics into a reversible framework using complex time, the GENERIC framework separates the reversible and irreversible contributions by introducing an explicit entropy variable. To compare the two frameworks, the differential equation of motion will be derived for the DHO using GENERIC. In this manner, the state variable for the DHO is defined as $z=\left(x,p,S\right)$, where the entropy S is also an independent variable, along with the position x and the momentum p. The total energy E is given as follows:(97)$$E\left(z\right)=\frac{p^{2}}{2m}+\frac{kx^{2}}{2}+u\left(S\right),$$
where $u\left(S\right)$ is the internal energy of the system with(98)$$\frac{\partial u}{\partial S}=\frac{1}{T}.$$The gradients of the energy E and entropy S with respect to the state variable z are calculated as follows:(99)$$\frac{\delta E}{\delta z}=\left(\begin{array}{c} kx\\ p/m\\ T \end{array}\right),\;\frac{\delta S}{\delta z}=\left(\begin{array}{c} 0\\ 0\\ 1 \end{array}\right).$$The Poisson operator $L\left(z\right)$ depends only on x and p, as follows:(100)$$L\left(z\right)=\left(\begin{array}{ccc} 0 & 1 & 0\\ -1 & 0 & 0\\ 0 & 0 & 0 \end{array}\right),$$Being antisymmetric, $L\left(z\right)$ satisfies L·∇S=0. Next, the dissipative operator $M\left(z\right)$ should be defined. It is important for $M\left(z\right)$ to contribute damping to $\dot{p}$, to preserve the total energy, and to increase entropy. Moreover, $M\left(z\right)$ must be symmetric, positive semi-definite, and satisfy M·∇E=0. Due to these reasons, the final form of $M\left(z\right)$ was determined as follows:(101)$$M\left(z\right)=b\left(\begin{array}{ccc} 0 & 0 & 0\\ 0 & 1 & -p/m\\ 0 & -p/m & {\left(p/m\right)}^{2} \end{array}\right).$$Using Equation (18), the final GENERIC structure for the DHO is written as follows:(102)$$\frac{dz}{dt}=\left(\begin{array}{ccc} 0 & 1 & 0\\ -1 & 0 & 0\\ 0 & 0 & 0 \end{array}\right)\cdot \left(\begin{array}{c} kx\\ p/m\\ T \end{array}\right)+b\left(\begin{array}{ccc} 0 & 0 & 0\\ 0 & 1 & -p/m\\ 0 & -p/m & p^{2}/\left(m^{2}T\right) \end{array}\right)\cdot \left(\begin{array}{c} 0\\ 0\\ 1 \end{array}\right).$$The above equation results in the derivation of the known equations of motion for the DHO, as follows:(103)$$\dot{x}=\frac{p}{m},\;\dot{p}=-kx-\alpha p,\;\dot{S}=\frac{\alpha }{mT}p^{2}.$$The GENERIC structure for the DHO highlights entropy as thermodynamic potential. It also ensures that the second law of thermodynamics $\dot{S}\geq 0$ holds, while the energy is conserved through M·∇E=0.

## 8. Discussion

In this paper, a Hamiltonian formulation is derived for the damped harmonic oscillator, both for the underdamped case and for the general form that includes all damping cases. The Hamiltonian is formed using Quantitative Geometrical Thermodynamics (QGT), originally developed by Parker and Jeynes [15,16,17,18,19]. This model is based on the complex time formalism, which allows one to reinterpret dissipation in classical systems through a unifying thermodynamic framework. By analytically continuing real time into the complex plane—where reversible dynamics evolves along the imaginary axis and irreversible processes along the real axis—one obtains a dual description of physical evolution in terms of energy (Hamiltonian) and entropy production. In this setting, the Hamiltonian of the system and its entropy production rate are not just analogous but mathematically isomorphic, connected through the Hilbert transform that emerges from causality. Such a result reframes the classical energy–entropy relationship as a deeper geometrical equivalence in the complex temporal domain.

Τhe derivation of the Hamiltonian for the DHO from entropy production was successfully achieved through QGT, validating and extending the formalism proposed by Parker and Jeynes [15]. Notably, for the underdamped case, the entropy production rate was shown to yield a real Hamiltonian of the form $U_{0}e^{\alpha t}$, matching McDonald’s relation (Equation (9)) [35] when $U_{0}$ is the average mechanical energy over a period. Due to the oscillatory nature of this case, the averaging produces a constant mechanical energy, which justifies the exponential form of the Hamiltonian. This confirms that dissipation, encoded in entropy production, can be Wick-rotated to reconstruct a valid Hamiltonian for systems with weak damping, where weak oscillations dominate.

However, extending this analysis to the general DHO solution, valid across all damping regimes, reveals a key distinction: the mechanical energy $U\left(t\right)$ is no longer constant but evolves explicitly in time, leading to a time-dependent Hamiltonian. Despite this, the formal structure of the Hamiltonian remains similar to the underdamped case, preserving the exponential factor $e^{\alpha t}$ and satisfying the same causality and Hilbert transform conditions prescribed by QGT. This verifies that QGT is not limited to near-conservative or weakly dissipative systems but is robust enough to derive well-defined Hamiltonians even for systems characterized by strong thermodynamic irreversibility, such as the critically damped and overdamped harmonic oscillators. Therefore, the generalized Hamiltonian derived from QGT not only recovers McDonald’s formulation [35] but unifies the regimes by embedding the entropy production directly into the Hamiltonian.

In general, the results of this work carry significant implications for our understanding of time asymmetry in dissipative systems. Classical mechanics, when formulated through conventional Hamiltonian dynamics, is inherently time-reversible, yet real-world systems ubiquitously exhibit irreversible behavior. By embedding entropy production directly into the Hamiltonian structure through the complexification of time, QGT offers a unified and elegant framework in which the arrow of time emerges not as an external constraint but as an intrinsic geometric feature of the system’s evolution. The appearance of imaginary energy components—manifesting as entropy production—demonstrates that the asymmetry between past and future is encoded within the analytical structure of the actio-entropy function itself. In this view, time irreversibility is not merely a statistical artifact but a fundamental consequence of the holomorphic extension of classical dynamics. Thus, the QGT formalism not only combines energy dissipation with Hamiltonian mechanics but also deepens our conceptual grasp of the origins of temporal directionality in physical systems.

It is noteworthy that the general Hamiltonian expression for the DHO derived using the QGT framework (Equation (93)) coincides in form with the Caldirola–Kanai (CK) Hamiltonian [30], but it is obtained via a completely different line of reasoning. In the CK formalism, the time-dependent exponential factor is introduced phenomenologically to ensure that the canonical equations reproduce the damped oscillator dynamics. On the other hand, the QGT derivation emerges from a geometric–thermodynamic foundation, in which the Hamiltonian and entropy production are related as Wick-rotated complex conjugates in a complexified time framework. The fact that these independent approaches converge to the same mathematical form indicates that the CK Hamiltonian may have a deeper geometric interpretation, and that QGT naturally captures it without phenomenological assumptions. This raises the possibility that the CK, Bateman, and other dissipative Hamiltonians may all be special cases of a broader QGT-based structure. Exploring this connection, along with its implications for both classical and quantum dissipative systems, represents a promising direction for future research.

An additional insight emerging from the QGT-derived Hamiltonian for the DHO is its compatibility with a generalized Poisson bracket structure, despite the presence of dissipation. Traditionally, Poisson brackets encode the symplectic geometry of conservative systems, where the Hamiltonian preserves phase-space volume (Liouville’s theorem). However, the exponential factor found in the QGT-derived Hamiltonian, derived from the entropy production, explicitly breaks time-reversal symmetry and induces phase-space conservation. Notably, by defining the canonical momentum in a manner similar to McDonald’s formulation [35], one can retain a modified yet meaningful Poisson bracket structure. In this formulation, the time evolution of the variables remains governed by Hamilton’s equations, but it now includes irreversible dynamics through the QGT Hamiltonian. Thus, the results of this paper enable the embedding of thermodynamically consistent dissipation into the framework of classical mechanics, laying a foundation for extension into metriplectic and GENERIC formulations that systematically unify reversible and irreversible dynamics.

A natural extension of this work lies in comparing the QGT-derived Hamiltonian structure with the GENERIC (General Equation for the Non-Equilibrium Reversible–Irreversible Coupling) framework. GENERIC provides a geometric structure that combines a Poisson bracket (antisymmetric) for reversible dynamics and a dissipative bracket (symmetric and positive semi-definite) for irreversible dynamics. In this setting, the state vector $z=\left(x,p,S\right)$—comprising position, momentum, and entropy—evolves according to the coupled flow equations generated by both the Hamiltonian and entropy. While QGT embeds dissipation through the complexification of time and derives a single effective Hamiltonian containing both reversible and irreversible content, GENERIC maintains a dual-generator structure where the Hamiltonian governs the reversible part and the entropy governs the irreversible. However, both approaches reveal an intimate link between entropy production and the time evolution of the dissipative system. It is evident from setting the QGT results side-by-side with GENERIC that the two frameworks align in their thermodynamic interpretation of dissipation, while they differ structurally, offering a bridge between complex-time Hamiltonians and metriplectic/GENERIC formulations.

Crucially, the QGT-derived Hamiltonian for the DHO integrates entropy production implicitly through its time dependence, matching the GENERIC requirement that the total energy is not conserved in the presence of entropy increase. While GENERIC preserves the explicit separation of structure via the $L\left(z\right)$ and $M\left(z\right)$ matrices (antisymmetric and symmetric, respectively), QGT internalizes these features into the analytic structure of the complexified Hamiltonian and its causal properties. Importantly, both frameworks converge on a consistent thermodynamic interpretation: entropy production is directly tied to irreversible evolution, whether captured as an explicit generator (in GENERIC) or as an imaginary energy component (in QGT). Therefore, the QGT result provides a minimal, unified Hamiltonian description of dissipation that enriches the more structured, bracket-based GENERIC approach. Table 1 summarizes the main conceptual and structural differences between the QGT formalism, classical Hamiltonian mechanics, and the GENERIC framework, highlighting how the present study applies and extends QGT to the damped harmonic oscillator.

Finally, it is important to note the isomorphism between the imaginary component of the complex time and the antisymmetric Poisson matrix through the lens of geometric algebra (see Appendix B). The imaginary component of complex time in QGT, responsible for reversible dynamics, plays the same structural role as the antisymmetric Poisson matrix L in GENERIC. Both define infinitesimal generators of rotation, preserve a conserved quantity (energy), and act on their respective spaces (complexified time and phase space) to produce reversible, unitary evolution. This correspondence allows us to interpret i and L as isomorphic in geometric algebra, suggesting that QGT’s complex geometry is a natural complexification of the real-valued symplectic geometry of conventional Hamiltonian mechanics. In this view, entropy production arises from the breakdown of symmetry between these two geometric structures—the real axis in QGT and the dissipative metric bracket in GENERIC.

## 9. Conclusions and Future Directions

In this work, it is demonstrated that a Hamiltonian relation for the damped harmonic oscillator (DHO) can be rigorously derived from entropy production using the previously established Quantitative Geometrical Thermodynamics (QGT) formalism. By treating time as a complex variable and interpreting entropy production as the imaginary component of energy, a unified Hamiltonian structure is recovered that remains valid across all damping regimes. This formulation not only reproduces the known results of McDonald’s Hamiltonian and Parker and Jeynes [15,35] for the underdamped oscillator, it also extends to critically damped and overdamped cases, embedding the full dissipative character of the system within a causally consistent, complex-analytic framework. Remarkably, the QGT-derived Hamiltonian was found to coincide in form with the well-known Caldirola–Kanai Hamiltonian, despite originating from fundamentally different principles. This nontrivial convergence reinforces the validity of the QGT approach and points to possible deeper geometric connections between time-dependent Hamiltonians and the entropy–energy duality in complex time. The QGT-derived Hamiltonian was also shown to preserve a generalized Poisson structure, while the imaginary time component appears to be isomorphic to the antisymmetric Poisson operator in GENERIC, further underscoring the intimate links among entropy, energy, and complex time.

The implications of these findings extend well beyond the scope of the DHO. First, the ability of QGT to encode irreversible thermodynamics in a single complex Hamiltonian suggests that similar approaches could be fruitfully applied to a broader class of dissipative systems, including quantum-damped harmonic oscillators, Navier–Stokes equations, reaction–diffusion systems, and viscoelastic media. Moreover, the geometric isomorphism between the imaginary time component and the Poisson matrix, as revealed through geometric algebra, opens a pathway to reinterpret classical mechanics in a holomorphic and geometrically unified formalism. Future work may explore how this framework interacts with spacetime algebra, providing insights into the arrow of time and causality at a relativistic or even cosmological scale. In quantum mechanics, the complex time approach may offer new routes to reconcile unitary evolution with decoherence and information loss, with potential connections to the thermodynamics of measurement, quantum entropy production, and Von Neumann entropy. Finally, the connection between energy and entropy production within complex time presents intriguing opportunities for advancing information theory, geometric thermodynamics, and classical mechanics, ultimately enriching our understanding of both microscopic and macroscopic irreversibility from a unified geometric standpoint.

## Figures and Tables

**Figure 1 entropy-27-00883-f001:**
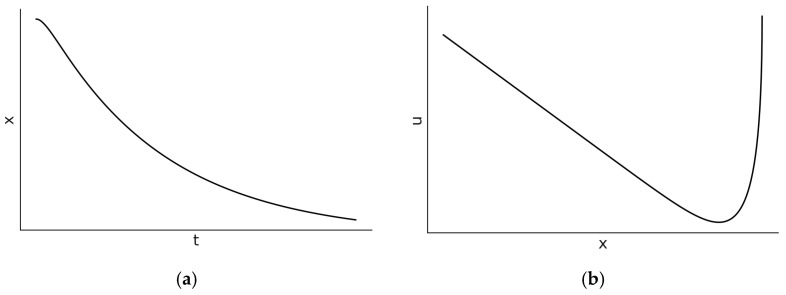
The (**a**) position–time dependence and (**b**) state-space diagrams (right side) for the overdamped harmonic oscillator where $\alpha^{2}>4\omega_{0}^{2}$. In Figure 1b, the velocity is denoted by u.

**Figure 2 entropy-27-00883-f002:**
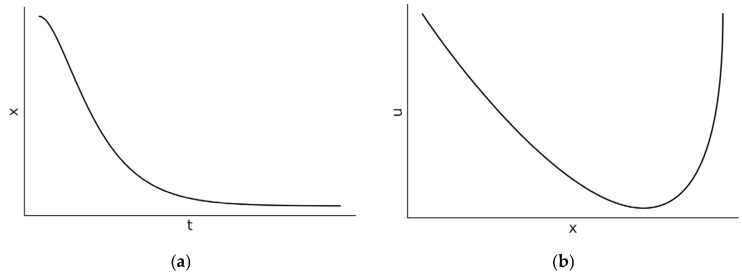
The (**a**) position–time dependence and (**b**) state-space diagrams (right side) for the critically damped harmonic oscillator where $\alpha^{2}=4\omega_{0}^{2}$. In Figure 2b, the velocity is denoted by u.

**Figure 3 entropy-27-00883-f003:**
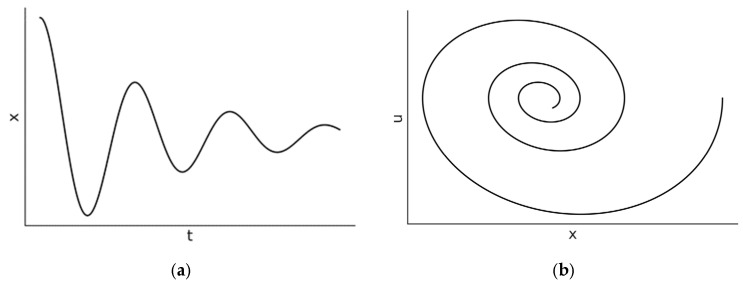
The (**a**) position–time dependence and (**b**) state-space diagrams (right side) for the underdamped harmonic oscillator where $\alpha^{2}<4\omega_{0}^{2}$. In Figure 3b, the velocity is denoted by u.

**Table 1 entropy-27-00883-t001:** Comparison among QGT formalism, classical Hamiltonian mechanics, and GENERIC for dissipative systems, such as the damped harmonic oscillator.

Feature	Classical Hamiltonian Mechanics	GENERIC	QGT (Parker & Jeynes, 2023 [15])
Role of time	Real time parameter; reversible dynamics only.	Real time; reversible and irreversible parts evolve concurrently but through distinct structures.	Complex time: real part gives entropy production (irreversible), imaginary part generates reversible evolution.
Treatment of dissipation	Cannot describe dissipative systems without modifications such as time-dependent Hamiltonians or auxiliary coordinates.	Dissipation treated using a symmetric bracket alongside reversible Poisson dynamics.	Dissipation embedded via complexification of time; entropy production enters directly into the Hamiltonian.
Entropy representation	Absent from canonical formalism.	Entropy, as a geometric object, is directly linked to dissipation.	Entropy is a scalar state function whose gradient drives the dissipative bracket, without complex time interpretation.
Poisson structure	Standard antisymmetric Poisson bracket preserving symplectic geometry.	Standard Poisson bracket with a symmetric dissipative bracket satisfying degeneracy conditions.	Modified Poisson bracket includes entropy-driven evolution while preserving geometric consistency.
Example: DHO Hamiltonian	Requires phenomenological forms (Caldirola–Kanai, Bateman dual).	Represents DHO by separating reversible and dissipative terms, requiring predefined entropy and friction operators.	Derived from entropy production; valid for all damping regimes.

## Data Availability

The data presented in this study are available upon request from the corresponding author.

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
