# Peer review of "Complex Time Approach to the Hamiltonian and the Entropy Production of the Damped Harmonic Oscillator"

_entropy, 2025, doi:10.3390/e27080883_

Round 1

Reviewer 1 Report

Comments and Suggestions for Authors

This is an interesting manuscript, in which the author proposes a novel thermodynamics-based method of the construction of the Hamiltonians of dissipative systems. Yet, I suggest that – prior the acceptance of the manuscript – the author should address several important issues detailed below.

  1. There are two standard methods of treating Hamiltonian dynamics of dissipative systems, by using Caldirola-Kanai Hamiltonian and through the Bateman dual system Hamiltonian. I suggest that the author should at least mention these methods and briefly discuss them.

  1. The author uses a thermodynamic (QGT) method for the construction of the Hamiltonian of the damped harmonic oscillator (HDO). In this method, temperature T enters the description in a standard thermodynamic way. For DHO, however, energy goes to zero at infinite time, meaning that T has to approach zero, too. What is the meaning of a finite T then?

  1. Related question. The conventional microscopic approach to thermodynamics is based on system-bath Hamiltonian methodology. Then thermodynamics of the system is constructed by integrating out the bath degrees of freedom. By doing that, we immediately realize that the system experiences not only energy dissipation (which is present in the DHO model), but also bath-induced fluctuations. This culminates in celebrated fluctuation-dissipation theorem, which ensures that the system will reach/maintain thermal equilibrium at temperature T. In DHO, we have dissipation only and thermal equilibrium is unachievable. Please comment on that.

  1. The already mentioned Caldirola-Kanai Hamiltonian is at odds with the Hamiltonian derived by the author (Eq. 93). Please explain & discuss.

  1. Can the proposed methodology be extended beyond the trivial DHO case?

Reviewer 2 Report

Comments and Suggestions for Authors

Round 2

Reviewer 1 Report

Comments and Suggestions for Authors

The author has adequately addressed my questions & concerns. I recommend publication in its present form.